# Suitability of Constitutive Models of the Structural Concrete Codes When Applied to Polyolefin Fibre Reinforced Concrete

**DOI:** 10.3390/ma15062323

**Published:** 2022-03-21

**Authors:** Alejandro Enfedaque, Fernando Suárez, Marcos G. Alberti, Jaime C. Gálvez

**Affiliations:** 1Departamento de Ingeniería Civil-Construcción, Universidad Politécnica de Madrid, E.T.S.I. Caminos, Canales y Puertos, 28040 Madrid, Spain; alejandro.enfedaque@upm.es (A.E.); marcos.garcia@upm.es (M.G.A.); 2Departamento de Ingeniería Mecánica y Minera, Universidad de Jaén, 23071 Jaén, Spain; fsuarez@ujaen.es

**Keywords:** fibre-reinforced concrete, polymeric fibres, Model Code 2010, cohesive model

## Abstract

The use of fibres as structural reinforcement in concrete is included in standards, providing guidelines to reproduce their behaviour, which have been proven adequate when steel fibres are used. Nevertheless, in recent years new materials, such as polyolefin fibres, have undergone significant development as concrete reinforcement. This work gives insight on how suitable the constitutive models proposed by the Model Code 2010 (MC2010) are in the case of such polymer fibres. A set of numerical models has been carried out to reproduce the material behaviour proposed by the MC2010 and the approach based on the softening function proposed by the authors. The results show remarkable differences between the experimental results and the numerical simulations when the constitutive models described in the MC2010 are employed for different polyolefin fibre reinforced concrete mixes, while the material behaviour can be reproduced with greater accuracy if the softening function proposed by the authors is employed when this type of macro-polymer fibres is used. Moreover, the relatively complex behaviour of polyolefin fibre reinforced concrete may be reproduced by using such constitutive model.

## 1. Introduction

Concrete has been one of the most employed construction material [1] in modern societies in spite of its impact on the environment [2]. Such use has been based on its resistance to natural weather conditions [3,4] and its remarkable compressive strength and modulus of elasticity [5,6]. However, plain concrete is also known as a quasi-brittle material which is not capable of sustaining high-tensile stresses. Adding reinforcing bars of concrete to form reinforced concrete was one of the possible solutions to this problem. Another possibility is adding fibres randomly distributed while mixing, forming fibre reinforced concrete (FRC). The effect of the addition of fibres in the mechanical properties of concrete depends on numerous parameters, such as the type of fibres used, dosage, shape and size, distribution, orientation and, among others, pull-out response [7,8]. Adding certain types of fibres in a determined volume fraction to concrete a material suitable for numerous applications can be obtained. For instance, steel fibre reinforced concrete (SFRC) has been successfully applied to pavements [9,10], tunnel linings [11,12], or even seismic structures [13,14]. However, the use of SFRC might have two main drawbacks, its durability under certain circumstances [15,16] and its environmental impact [17]. One possibility to overcome such durability issues is to use new polymer macro-structural fibres, which are chemically stable when applied in foundations, tunnel linings or marine environments [18,19,20,21]. Polyolefin fibre reinforced concrete (PFRC) is one of the materials available for these uses.

All the previous applications have been possible due to several national codes and recommendations [22,23,24,25,26,27]. These documents have set the conditions that a certain FRC should meet so that the improvements provided by the fibres added might enable a reduction in the amount of steel reinforcing bars used. Such requirements take into account the residual strength of the FRC studied at certain crack openings which are related to the service limit state (SLS) and the ultimate limit state (ULS). For checking if a FRC formulation boasts sufficient residual strength, bending tests are performed on FRC specimens in laboratory conditions. Once such tests have been performed, and if the FRC properties exceed the requirements set, the contribution of the fibres might be included in the structural design of the concrete element. Although it is true that both SFRC and PFRC have been shown as suitable for their application in structural concrete elements [7,28,29] it should not be overlooked that the extensive experience related with SFRC served as reference when the requirements of the material were set. Once such requirements have been checked, the aforementioned recommendations propose several constitutive models that should be applied to the structural design and that have been successfully applied when using SFRC. Similarly, the constitutive models proposed used several characteristics that were inspired by the mechanical behaviour of SFRC. Consequently, the accuracy of such approaches when dealing with PFRC is a matter that deserves being studied.

The behaviour of PFRC can meet the requirements in the standards [30] although the shapes of the fracture curves have different features compared with those of SFRC [29]. According to the Model Code 2010 (MC2010) [31], the first condition that FRC should meet is a minimum load value, proportional to the maximum strength in the limit of proportionality, at a crack mouth opening displacement of 0.5 mm; this limitation is intended to prevent brittle behaviour of the material. Although PFRC with an average fibre dosage might meet the cited requirement, polyolefin fibres need greater deformations to absorb the load that concrete is not capable of sustaining being the first unloading branch steeper than in SFRC. The second condition that MC2010 establishes is that at a crack mouth opening displacement of 2.5 mm the load must be over 20% of the strength at the limit of proportionality. This limitation for PFRC is met even for very low fibre dosages [30]. As there are remarkable differences in the post peak mechanical behaviours of SFRC and PFRC, the applicability of the constitutive models, which are built based on certain values of their residual strength, should be studied. Thus, this study seeks to address the main issues concerning the considerations in the standards regarding the mechanical behaviour of PFRC.

The main goals of this contribution are the following. First, an implementation of the linear and rigid-plastic models developed in MC2010 in a numerical code by means of a material subroutine is sought. Second, the differences between the material behaviour obtained with the constitutive models of MC2010 and those obtained using the trilinear approach proposed by the authors that delivers an accurate reproduction of the PFRC behaviour are assessed. This matter will be carried out in four mixes of PFRC with 3, 4.5, 6, and 10 kg/m^3^ of polyolefin fibres. Lastly, an evaluation of the material behaviour at SLS and ULS and another regarding the fracture energy consumed in the tests are carried out for determining the reliability of using the studied approaches. Summarising, this manuscript supplies additional information for a possible future adaptation of MC2010, and other national concrete structural codes based on it when applied to PFRC.

## 2. Post-Peak Material Modelling

In this section, the numerical modelling of fracture used in the simulations is addressed. In the first part, the embedded cohesive crack model is briefly presented. Since the reader can find a detailed description in previous works published by the authors, this description has been kept to a minimum. In the second part, the model is adapted for PFRC following the guidelines of the MC2010.

### 2.1. Embedded Cohesive Crack Modeling

Fracture is modelled by means of a cohesive crack model, based on the work proposed by Hillerborg [32] and implemented as an embedded cohesive crack developed in a triangular finite element. This numerical modelling has been used for simulating fracture of concrete [33] and has been adapted for use with brick masonry [34] and, more recently, with PFRC [35].

In this model, fracture is assumed to develop mainly under mode I and with the cohesive stress vector t being parallel to the displacement vector w (thus, it is a central force model), as expressed by (1):(1)$$t=\frac{f\left(\left|\tilde{w}\right|\right)}{\left|\tilde{w}\right|}w\;\;\;\;\;\;\;\;\;\;\;\;\;\;\;\;\;\;\;\;\;\;\;\;\;with\;\tilde{w}=max\left(\left|w\right|\right)$$
with $f\left(\tilde{w}\right)$ being the softening function, expressed in terms of $\tilde{w}$, which corresponds to an equivalent crack opening that considers the maximum historical opening, necessary for correctly reproducing possible loading-unloading situations.

This formulation is implemented for triangular elements and, as cracks can only develop in lines parallel to the sides of the element, only three cracking orientations are allowed (see Figure 1). It follows a strong discontinuity approach, dividing the element into two parts, A^+^ and A^−^, with t being parallel to the crack displacement and constant along the length of the crack L. It is obtained by (2):(2)$$t=\frac{A}{hL}\sigma \cdot n\;$$
with A being the area of the element, h the height of the triangle over the side opposite to the solitary node and n the unit vector normal to the crack (see Figure 1).

The material around the crack is elastic and the stress tensor is obtained by using a strain vector that results from subtracting the strain due to the crack displacement from the apparent strain $\varepsilon^{a}$, obtained under elastic considerations and by using the nodal displacements (Equation (3)):(3)$$\sigma =E:\left[\varepsilon^{a}-{\left(b^{+}⨂w\right)}^{S}\right]$$
where E stands for the elastic tangent tensor, $\varepsilon^{a}$ for the apparent strain vector obtained with the nodal displacements, $b^{+}$ for the gradient vector of the shape function of the solitary node, which can be obtained as:(4)$$b^{+}=\frac{1}{h}n$$

Superscript *S* makes reference to the symmetric part of the tensor, the usual double-dot product (${\left(A:B\right)}_{ij}=A_{ijkl}b_{kl}$) and ⨂ the usual direct product (${\left(a\;⨂b\right)}_{ij}=a_{i}b_{j}$).

Lastly, by using the expression of the stress tensor (3), the stress vector can be obtained as:(5)$$t=\frac{f\left(\left|\tilde{w}\right|\right)}{\left|\tilde{w}\right|}w=\left[E:\varepsilon^{a}\right]\;\cdot \;n-\left[E:{\left(b^{+}⨂w\right)}^{S}\right]n$$

This can be rewritten as:(6)$$\left[\frac{f\left(\tilde{w}\right)}{\left|\tilde{w}\right|}1+n\cdot E\cdot b^{+}\right]\cdot w=\left[E:\varepsilon^{\alpha }\right]\;\cdot \;n$$
where 1 identifies the second-order identity tensor.

By using this expression, the crack displacement can be solved for a certain set of nodal displacements by means of the Newton–Raphson method. This model has been implemented by using a user subroutine UMAT of ABAQUS^®^ and, since it needs information about the geometry of the model (the nodes coordinates of each element), it reads a previously prepared external file.

### 2.2. Model Adopted by the MC2010

The MC2010 provides two possible constitutive laws to reproduce the behaviour of FRC, with one describing a rigid-plastic behaviour and another one a linear behaviour. These models are defined by parameters that make reference to the values of the residual flexural tensile strength and the crack mouth opening displacement (CMOD) at specific instants during a three-point bending test. In particular, the expression of the residual flexural tensile strength for a specific value of the CMOD may be obtained by expression (7):(7)$$f_{R,j}=\frac{3F_{j}l}{2bh_{sp}^{2}}$$
where $f_{R,j}$ is the residual flexural tensile strength in MPa corresponding to a value of the $CMOD=CMOD_{j}$ in mm, $F_{j}$ is the load in N applied when $CMOD=CMOD_{j}$ (in mm), l is the span length, b the specimen width and $h_{sp}$ the height of the notched section (ligament) where the crack propagates, with all these dimensions being in mm (see Figure 2b). Consequently, $f_{R1}$ stands for the residual strength at a CMOD of 0.5 mm and $f_{R3}$ stands for the residual strength at a CMOD of 2.5 mm. Figure 2a shows a typical load-CMOD curve for FRC where some values of F and CMOD are identified according to the previous description, as described in the MC2010. Expression (7) can be easily obtained by applying the basic principles of the strength of materials at the notched section of the specimen.

As mentioned before, MC2010 proposes two possible constitutive laws for FRC. The rigid-plastic model simply describes a constant stress behaviour from the point at which cracking begins (w=0) until it reaches a limit value ($w_{u}$). In this model, the constant stress is obtained by considering the ultimate state when $CMOD=CMOD_{3}$ and the ultimate value of w is considered as $w_{u}=CMOD_{3}$.
(8)$$f_{Ftu}=\frac{f_{R3}}{3}$$

The linear model is defined by two reference values, $f_{Fts}$ and $f_{Ftu}$, which can be obtained by (9) and (10):(9)$$f_{Fts}=0.45f_{R1}$$
(10)$$f_{Ftu}=f_{Fts}-\frac{w_{u}}{CMOD_{3}}\left(f_{Fts}-0.5f_{R3}+0.2f_{R1}\right)\;\geq \;0$$

These values define a line which, as can be observed in these equations, is based on the experimental values for $CMOD_{1}$ and $CMOD_{3}$. Neverheless, the ultimate crack opening *w_u_* is not specifically defined and is kept as the maximum crack opening accepted in structural design, which depends on the ductility required.

Figure 3 shows the schemes for both models: the rigid-plastic and the linear model. It should be noted that while in this figure the linear model shows a load-descending behaviour, depending on the performance of the PFRC, the linear model may present a positive slope which in the MC2010 is referred to as post-crack hardening behaviour.

### 2.3. Trilinear Model Proposed for PFRC

In past papers published by the authors, the behaviour of hardened specimens of self-compacting concrete (SCC) and vibrated conventional concrete (VCC) was studied, with particular examination of how the increase of fibre dosages affects the flexural behaviour of specimens under three-point bending tests [35]. The authors proposed the use of a trilinear softening curve able to reproduce the classical three-point bending tests that develops under pure mode I conditions, as well as a modified version of the test proposed in [36], where cracking is initiated under a combination of modes (I) and (II) [37].

This trilinear model can be described by three linear parts, as can be seen in Figure 4. The first linear part reproduces the load decrease due to the failure of the concrete matrix up to the instant at which fibres make a relevant contribution to the specimen flexural strength. The second linear part reproduces the contribution of the fibres up to their maximum tensile capacity and in the third part the load decreases due to the final failure of fibres, which can be due to fibre tensile failure or because of the slip of fibres inside the concrete.

### 2.4. Stress-Crack Opening Diagrams Used in This Study

In order to compare the suitability of each of the three approaches described above, this study examines the experimental behaviour of four PFRC mixes with the numerical predictions of each of them.

The reference experimental results correspond to four PFRC mixes of VCC with four fibre dosages: 3, 4.5, 6 and 10 kg/m^3^. These are selected results from a broader experimental campaign that studied several PFRC mixes and analysed their mechanical properties through a series of two prismatic and nine cylindrical specimens of each mix; the details regarding these experimental works may be consulted in [28,29].

Regarding the constitutive laws defined by the MC2010, which have been described before, Table 1 shows the parameters for each fibre dosage and for each of the two possibilities described in the standard: rigid-plastic model and linear model. Note that the linear model is defined by two points: one for w=0 (defined by) and another one defined by the value of $w={\mathrm{CMOD}}_{3}$. These diagrams are shown in Figure 5 and have been defined following the MC2010, thus a limit value of the crack opening of 2.5 mm has been employed and Equations (8)–(10) used with the values of *f_R1_* and *f_R3_* experimentally obtained, which can be consulted in [28,29].

As for the trilinear model, Table 2 shows the coordinates of the points that define them for each fibre dosage. The inverse analysis performed to obtain these values may be consulted in [35].

## 3. Numerical Simulations

Numerical simulations were carried out by means of the finite element method with the commercial software ABAQUS^®^ (version 6.18). The softening behaviour was reproduced by the embedded cohesive crack formulation described above, implemented through a user subroutine UMAT in ABAQUS^®^. Since, as required by the mathematical formulation described before, only linear triangular elements have been used, and the mesh is refined along the line where cracking develops (see Figure 6a); this refinement was designed based on previous works (see [37,38]) where the mesh dependence was already analysed.

In the case of models where fracture is reproduced by a trilinear model and by the linear model of the MC2010 with softening (negative slope of the diagram), all the elements have been modelled by using the same cohesive formulation as that presented before. In the case of those constitutive models with abrupt decrease of strength, like the rigid-plastic model and that termed “linear model with limited development”, convergence issues are usual when employing an implicit integration scheme. The convergence problem in the models that uses “linear softening with unlimited development” with hardening (positive slope of the diagram) provides a different explanation. In these cases, the material becomes harder with increasing values of w, (if an element is damaged it becomes harder, meaning that damage is more likely to develop around it instead of progressing in the element that is already damaged). This leads to an increasing wider region of the specimen that develops, with it being more difficult for the model to find the crack path since, as load displacement increases, there are more possible elements to develop damage which means that convergence is not reached. Figure 6b shows one of these models where damage started developing along the expected crack path but, at some point, started to develop around it and finally convergence was not reached when there were many possible elements to develop damage.

In such cases with convergence problems, the path of fracture was tracked by defining elements that could be damaged only in the crack path, with the rest of elements behaving elastically.

### Results

Figure 7 shows the load-deflection curves for each of the approaches described for various fibre dosages: 3, 4.5, 6 and 10 kg/m^3^, therefore reproducing the experimental results of [28,29].

It is interesting to observe the differences among these approaches. In the case of the rigid-plastic model, diagrams always show a plateau and a subsequent load decrease that describes an exponential-like curve with small load jumps. As these drops are due to the abrupt load decrease described by the rectangular diagram, when an element reaches $w=w_{u}$, there is a sudden load decrease.

In the case of the “linear model with unlimited development”, the load-deflection diagrams exhibit a linear behaviour after the concrete matrix fails and, depending on the fibre dosage the specimen, may show a softening behaviour (for 3 and 4.5 kg/m^3^) or a hardening behaviour (for 6 and 10 kg/m^3^). Obviously, in the case of diagrams with hardening, this behaviour cannot be realistic, since the load may increase without a limit. Nevertheless, it is interesting to compare these results with those of the linear model with limited development that display an analogous behaviour until the most damaged element reaches $w=w_{u}$, when the load-deflection diagrams show a highly similar behaviour to that observed with the rigid-plastic model. In all cases, the load drop follows an exponential-like shape with small jumps that can be explained as for the rigid-plastic model.

Lastly, if the results obtained with a trilinear model are observed, remarkable differences can be highlighted compared with any of the results obtained with the other approaches. First, the maximum load reached that identifies the beginning of the fracture process is higher than in any of the other cases and the shape shows an initial load decay, followed by a load recovery and another load decay later. Second, the last load decay, which identifies the last part of the test until failure, shows a curved shape that has the opposite curvature if compared with results obtained with the rigid-plastic model and the linear model with limited development.

## 4. Discussion

In Figure 8 it can be seen that in the case of PFRC3 there are remarkable changes in the shape of the curves obtained through using the post-peak constitutive functions analysed. Regarding the rigid-plastic model and the linear model proposed by the MC2010, none is capable of reproducing the peak load registered in the experimental tests (it should be taken into account that usually 3 kg/m^3^ of polyolefin fibres do not allow PFRC to be classified as structural concrete). However, the trilinear approach accurately captures such a peak load. This phenomenon might be explained because the trilinear approach is the only one that establishes a progressive unloading after reaching peak load. Similarly, the trilinear approach is able to simulate the unloading process of the PFRC3 formulation. In the case of the rigid-plastic approach, when the deflection increases the load-deflection curve features a plateau. This stretch ends when a deflection of 3 mm is reached. From that deflection onwards, a progressive unloading takes place which does not fit the experimental behaviour. In the case of the “linear approach (unlimited)”, as the slope is negative the load decreases as the deflection increases. Consequently, such simulation is able to maintain its load-bearing capacity beyond 6 mm of deflection. The results obtained with the “linear model (limited)” softening curve also boast a descending slope until a progressive unloading starts at 3 mm of deflection. From that point onwards, the curve is not capable of reproducing the experimental behaviour. It should be underlined that the rigid-plastic approach and both linear approaches obtain load–deflection curves that are above the experimental one until a deflection of 2.5 mm is reached. For greater deflections, the experimental curve is above that predicted by both linear approaches. In the case of the rigid-plastic model, a similar comment could be made. However, in the case of the trilinear approach a close reproduction of the experimental behaviour PFRC3 is achieved throughout the test.

In the part of Figure 8 that corresponds to PFRC4.5, it may be observed that the differences perceived in the curves related with PFRC3 still appear. The peak load, as in the previous case, is not captured by any approach proposed by the MC2010 or even the modification proposed for the linear model. However, the trilinear constitutive model is able to simulate the flexural tensile behaviour of the material prior to the development of the failure surface. As in the case of PFRC3, the trilinear softening function was able to simulate the experimental behaviour of the PFRC4.5 in all its stages. Regarding the behaviour of the rigid-plastic model, it may be seen that its main feature is a constant load-bearing capacity until a 3 mm deflection is reached. From that point onwards the rigid-plastic model is below the experimental curve. As regards to both linear approaches are above the experimental curve between 0.5 and 2.5 mm of deflection. There are few differences between the rigid-plastic approach and the linear approach because the slope of the linear model is close to zero. One feature that could be highlighted is that the unlimited linear model behaviour is close to the experimental one for deflections between 2.5 and 6 mm.

Regarding the curves shown for PFRC6, a similar trend may be perceived if the experimental curve is compared with the one obtained with the rigid-plastic approach. For deflections ranging from 0.3 mm to 2.5 mm, the simulated curve is above the experimental one as the plateau is close to the experimental maximum post-peak load. From 2.5 mm of deflection, the unloading branch of the rigid-plastic model features a steep degree of unloading that is below the experimental behaviour. Regarding the linear model with the unlimited option, it should be pointed out that as the slope of the load-deflection curve is positive, the model is not able to capture the material failure as its load bearing capacity increases with deflection. Thus, beyond 3.5 mm of deflection the linear model is above the experimental curve. This also occurs between 0.3 mm and 2.5 mm of deflection. If the limited linear model is studied, similar comments to those previously mentioned regarding its unlimited version could be made until 2.5 mm of deflection is reached. From greater deflection values there are remarkable differences between the behaviour of the model and the real one because steep unloading appears in the simulated curve. As regards the simulation performed with the trilinear approach, it could be said that it is able to reproduce the shape and values of the experimental curves although there are certain differences from 3.5 mm onwards. In this case, the experimental behaviour is slightly above the simulated one.

Regarding PFRC10, similar features to those observed in PFRC6 can be seen. The rigid-plastic approach has similar characteristics than those previously mentioned in the case of PFRC6. A similar trend may be mentioned when the differences between the experimental curve and both linear approaches are evaluated. On this occasion, the slope of the linear model is positive, and no failure of the material occurs. If the model is limited a under evaluation of the behaviour of PFRC10 is observed. Regarding the trilinear approach, for deflection values greater than the maximum post-peak load such a curve marginally underestimates the load-bearing capacity of the material.

Figure 9 provides a comparison among the loads registered at different deflection values in the simulations performed and takes into account the three constitutive models studied. The values shown are relative to the value obtained in the experimental test which is considered as unity. Consequently, if a load value obtained in the simulation is greater than the one obtained in the test, the column that appears in Figure 9 is greater than one. On the contrary, values lower than the unity reflect that the simulation underestimates the real load-bearing capacity of the material. Such load analysis has been performed at three deflections: 0.5 mm, 2.5 mm and 5 mm. The values of deflection studied are based on the almost linear relation that exists between the crack width and the deflection. Consequently, it may be considered that a deflection value of 0.5 mm is close to the crack width used for checking the SLS of the structural element. Correspondingly, the deflection value of 2.5 mm has been chosen because such a value is close to the crack width chosen when analysing the ultimate limit state of the structural element. Lastly, regarding the 5 mm of deflection, such a value has been chosen as it may reflect the suitability of the constitutive models to detect the ultimate bearing capacity of the material at great deflections.

When the analysis of the SLS is performed for the PFRC formulations, several relevant features are evident. In the case of PFRC3, it may be seen that while the trilinear approach underestimates the capacity of the material, the two proposals of the MC2010 boast loads greater than the experimental one. In the case of the linear model, both versions obtain a load approximately 1.6 times the experimental one. In the case of the rigid-plastic model, although it also overestimates the experimental value registered, it is much closer to the real one. In the case of PFRC4.5, all the values obtained in the simulations are greater than the experimental ones (except the trilinear one which is much closer to the real value). On this occasion, the load values of the rigid-plastic model are close to those obtained with the linear models. Similar results were obtained for the PFRC6 and PFRC10 formulations.

These features seem to be a consequence of the load-crack mouth opening displacement curve taken as reference to set the requirements needed for considering the contribution of fibres in FRC (see Figure 2a). As such a reference curve boasted an $f_{R3}$ value lower than fR1 if the design values are obtained by reducing the load obtained in $f_{R3}$, such a constitutive model might have an accurate correlation with material behaviour. Nevertheless, in the case of PFRC, given that the value of $f_{R3}$ is greater than $f_{R1}$, the design value obtained for reduced strains might not be an accurate reproduction of the experimental behaviour.

When analysing the ULS, similar to the behaviour that appears at 2.5 mm of deflection, it may be seen that the trilinear approach and those proposed by the MC2010 reflect a notable degree of accuracy with the experimental behaviour of the material.

Regarding the loads obtained with the trilinear approach, all load values are within a 20% margin from the experimental at 5 mm of deflection. In the case of the rigid-plastic model and the limited linear model, both are far from the experimental results for any fibre dosage. In the case of the unlimited linear model, the load values are below the experimental ones when the slope is negative. On the contrary, when the slope is positive such values are slightly above those registered in the tests.

Another material property that might be of interest is the fracture energy required to generate certain deflections of the concrete element. Consequently, the fracture energy at the aforementioned deflections has been obtained for all the concrete formulations and constitutive models analysed. Those results can be seen in Table 3, where the variation of the fracture energy of the curves obtained in the simulations performed by using the constitutive models implemented has been related to that obtained in the experimental tests.

In the case of the trilinear model, it can be seen that a notable precision was obtained at all the deflections analysed. Regarding the rigid-plastic model, there was a certain degree of scattering between the values obtained at 0.5 mm between the experimental values and the simulated ones. Such scattering was reduced at 2.5 mm of deflection and increased at 5 mm of deflection. In the case of the linear model, the accuracy of the prediction of the fracture energy absorbed by the material increased as the deflection grew. In Table 3 it may be seen that the proposed limitation of the linear approach did not enhance the accuracy of the prediction of the fracture energy absorbed.

## 5. Conclusions

The present contribution has implemented the constitutive relations proposed by the MC2010 for FRC in a user material subroutine seeking to assess the adaptability of these constitutive relations to PFRC. In addition, a trilinear model and an unlimited version of the linear model proposed by MC2010 have been employed as comparisons.

The simulations carried out with the implementation of the trilinear softening function proposed by the authors showed a remarkable resemblance with the experimental behaviour of PFRC. Regarding the simulations performed through using the constitutive models proposed by MC2010, it was observed that they might not replicate the flexural behaviour of PFRC precisely. Neither the peak load nor the unloading process appeared in the simulations performed with the rigid-plastic and linear constitutive relations.

When analysing the load values obtained at 0.5 mm of deflection, it is seen that the constitutive relations proposed by the MC2010 are above the experimental results. At ULS, the relations proposed by the MC2010 and that proposed by the authors are able to reproduce with notable exactness the experimental load values. Regarding the ultimate load-bearing capacity of the material, which was assumed at a deflection of 5 mm, both the trilinear and unlimited linear model were close to the load-bearing capacity of the material. On the contrary, the rigid-plastic model and the limited version of the linear model clearly underestimated the mechanical properties of PFRC.

Based on the aforementioned arguments, it seems that the rigid-plastic and the linear models proposed by the MC2010 might be employed with caution at SLS when applied to PFRC. This might be a result of the limited degree of accuracy that the idealisation considered in MC2010 may offer when applied to FRC formulations where *f_R3_* is notably greater than *f_R1_*. The authors consider that if the results of the EN-14651 fracture tests boast a clear three-stretch behaviour after reaching the peak load (unloading-reloading-final unloading), a trilinear constitutive relation might be adopted as an alternative to the models proposed by MC2010. This is relevant given that the latter models were developed considering that steel fibres were the main option for reinforcing concrete with fibres.

The limitation of this study might be that the differences detected were determined for certain constitutive models stated in MC2010 and consequently if such models are modified the present conclusions should be re-evaluated.

## Figures and Tables

**Figure 1 materials-15-02323-f001:**
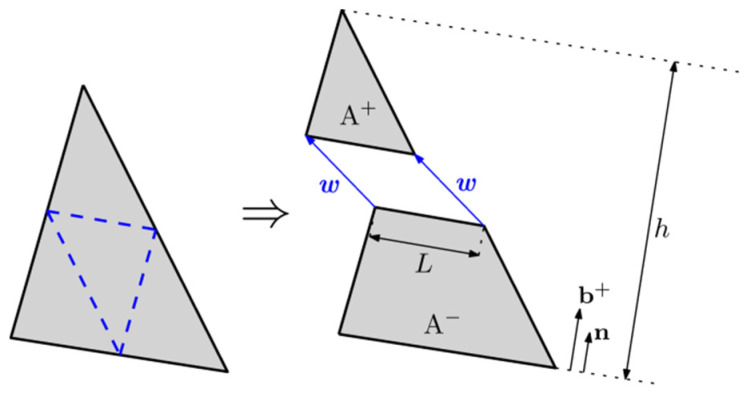
Possible crack paths (**left**) and geometrical definitions of w, n and $b^{+}$ (**right**).

**Figure 2 materials-15-02323-f002:**
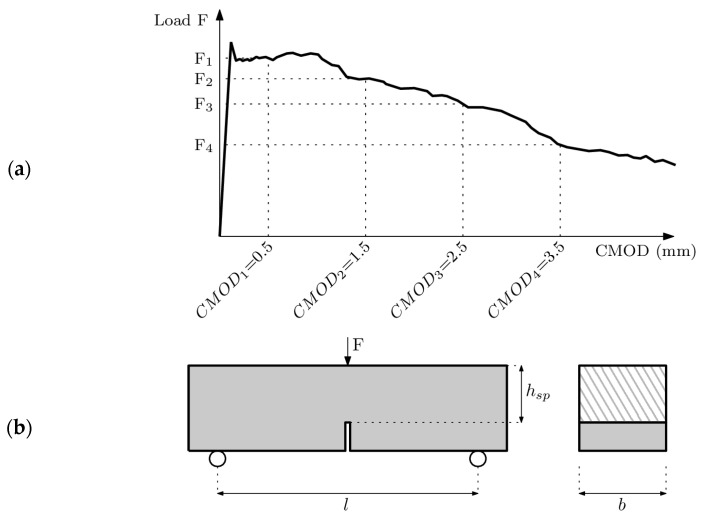
(**a**) Typical load F-CMOD curve for FRC and (**b**) geometry of a three-point bending test specimen.

**Figure 3 materials-15-02323-f003:**
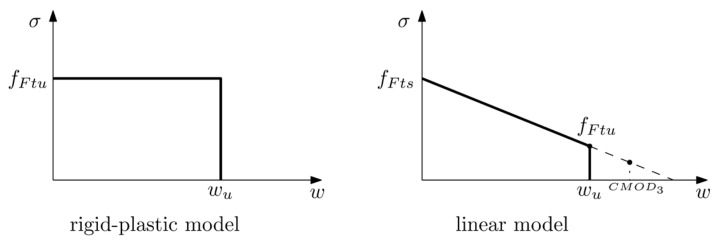
Rigid-plastic model (**left**) and linear model (**right**) as described by the MC2010.

**Figure 4 materials-15-02323-f004:**
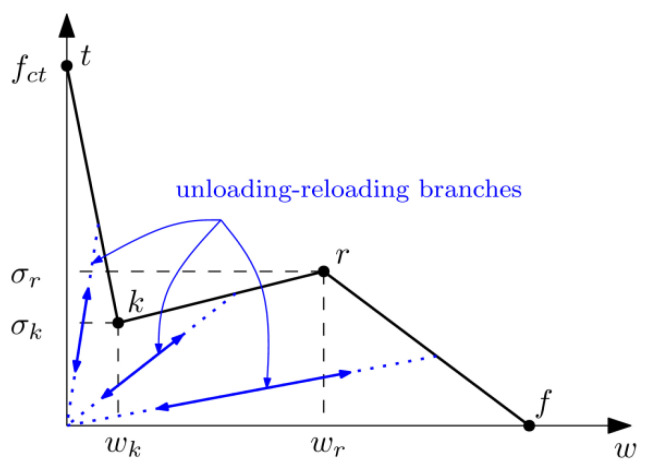
Trilinear diagram proposed by the authors for PFRC.

**Figure 5 materials-15-02323-f005:**
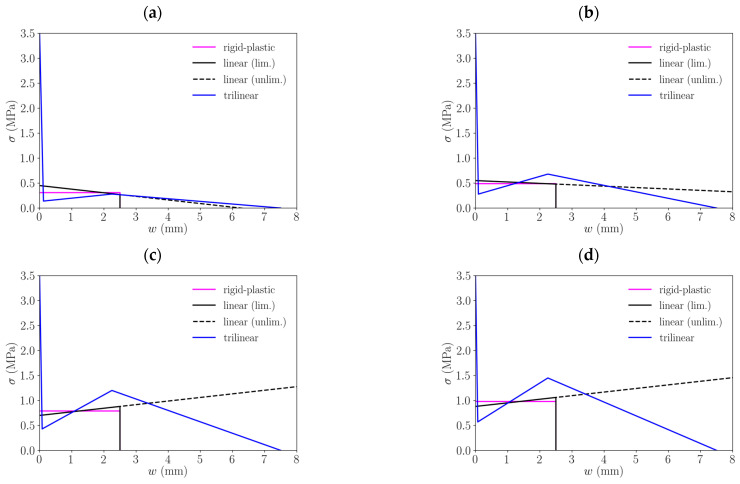
Comparison of the constitutive models obtained by each approach for different fibre dosages: (**a**) 3 kg/m^3^, (**b**) 4.5 kg/m^3^, (**c**) 6 kg/m^3^, (**d**) 10 kg/m^3^.

**Figure 6 materials-15-02323-f006:**
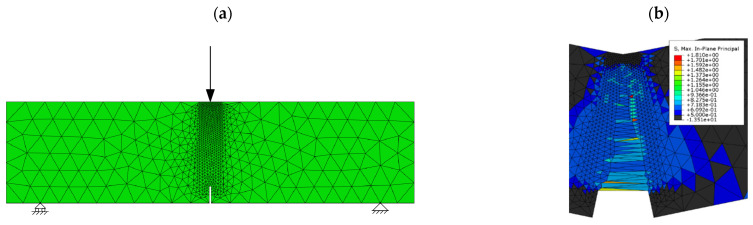
(**a**) Mesh refined along the crack path and boundary conditions, (**b**) maximum principal stress field in the damaged region of the model that used the linear constitutive law for 6 kg/m^3^ fibre dosage with no crack tracking. Convergence is not reached.

**Figure 7 materials-15-02323-f007:**
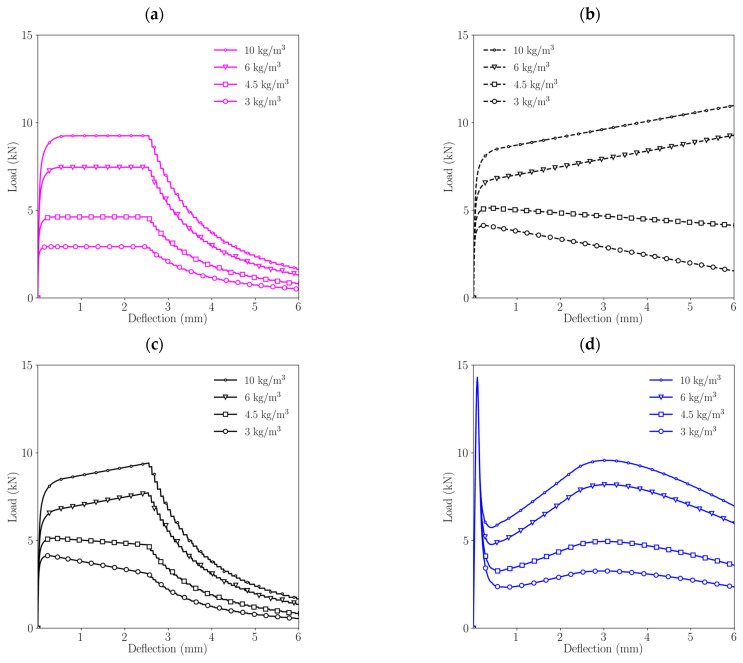
Comparison of the load-deflection diagram obtained with each approach for different fibre dosages: (**a**) rigid-plastic model, (**b**) linear elastic model (unlimited development), (**c**) linear elastic model (limited development), (**d**) trilinear model.

**Figure 8 materials-15-02323-f008:**
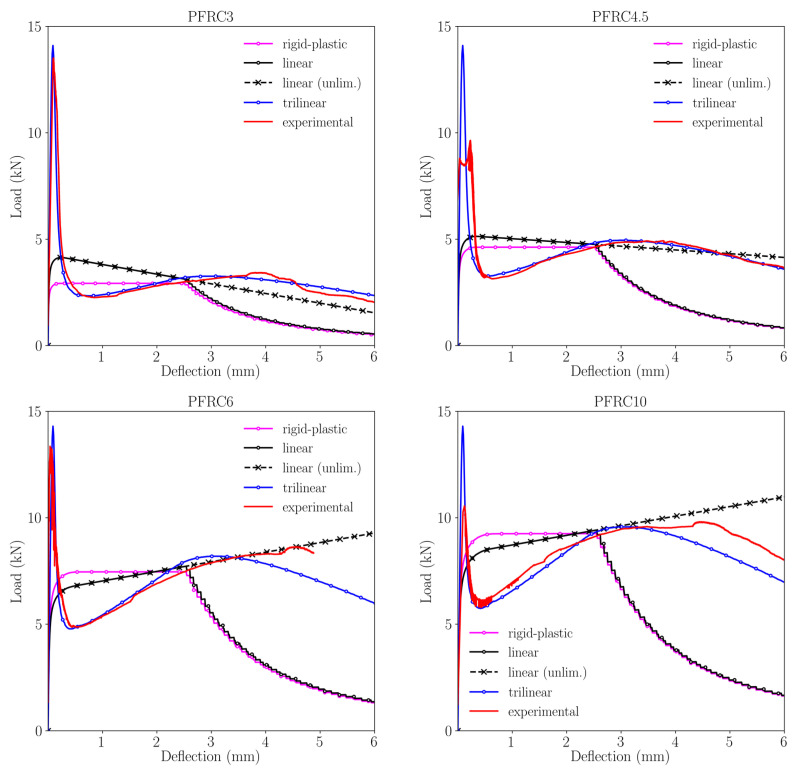
Comparison among the experimental curves and the simulations results obtained with the constitutive models implemented.

**Figure 9 materials-15-02323-f009:**
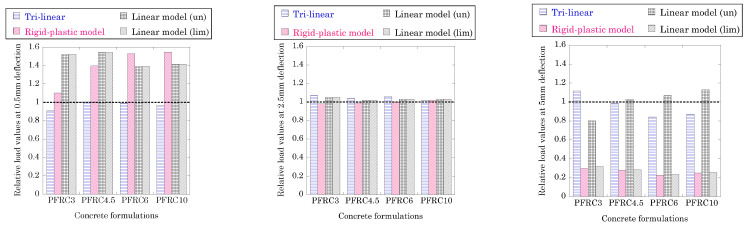
Comparison of the loads borne by the simulations at certain deflection values with respect to the experimental values.

**Table 1 materials-15-02323-t001:** Parameters of the rigid-plastic and the linear models for each fibre dosage of PFRC.

	Rigid-Plastic Model		Linear Model		
	$f_{Ftu}(\mathrm{MPa})$	$w_{u}(\mathrm{mm})$	$f_{Fts}(\mathrm{MPa})$	$f_{Ftu}(\mathrm{MPa})$	$w_{u}(\mathrm{mm})$
PFRC3	0.31	2.5	0.45	0.27	2.5
PFRC4.5	0.49	2.5	0.55	0.48	2.5
PFRC6	0.79	2.5	0.70	0.88	2.5
PFRC10	0.98	2.5	0.88	1.06	2.5

**Table 2 materials-15-02323-t002:** Coordinates of the points that define the trilinear model for each fibre dosage according to the point nomenclature provided in Figure 3.

	T		k		r		f	
	σ (MPa)	*w* (mm)	σ (MPa)	*w* (mm)	σ (MPa)	*w* (mm)	σ (MPa)	*w* (mm)
PFRC3	3.5	0	0.14	0.12	0.28	2.25	0	7.5
PFRC4.5	3.5	0	0.28	0.09	0.68	2.25	0	7.5
PFRC6	3.5	0	0.43	0.08	1.20	2.25	0	7.5
PFRC10	3.5	0	0.57	0.07	1.45	2.25	0	7.5

**Table 3 materials-15-02323-t003:** Fracture energy consumption in the experimental tests and in the simulations.

	Experimental	Trilinear		Rigid-Plastic Model		Linear Model (un.)		Linear Model (Lim.)	
PFRC3	*G_F_* (N/mm)	*G_F_* (N/mm)	∆ (%)	*G_F_* (N/mm)	∆(%)	*G_F_* (N/mm)	∆ (%)	*G_F_* (N/mm)	∆ (%)
0.5 mm	0.160	0.147	−7.99	0.076	−52.55	0.106	−33.47	0.106	−33.47
2.5 mm	0.435	0.433	−0.60	0.387	−11.07	0.486	11.63	0.486	11.63
5 mm	0.849	0.846	−0.28	0.585	−31.07	0.829	−2.31	0.698	−17.75
PFRC4.5	*G_F_* (N/mm)	*G_F_* (N/mm)	∆ (%)	*G_F_* (N/mm)	∆ (%)	*G_F_* (N/mm)	∆ (%)	*G_F_* (N/mm)	∆ (%)
0.5 mm	0.199	0.160	−19.68	0.118	−40.63	0.131	−34.23	0.131	−34.23
2.5 mm	0.609	0.582	−4.47	0.610	0.14	0.657	7.88	0.657	7.88
5 mm	1.239	1.210	−2.34	0.924	−25.43	1.262	1.83	0.978	−21.06
PFRC6	*G_F_* (N/mm)	*G_F_* (N/mm)	∆ (%)	*G_F_* (N/mm)	∆ (%)	*G_F_* (N/mm)	∆ (%)	*G_F_* (N/mm)	∆ (%)
0.5 mm	0.196	0.176	−9.91	0.181	−7.42	0.164	−16.04	0.164	−16.04
2.5 mm	0.849	0.849	−0.04	0.979	15.34	0.941	10.88	0.941	10.88
5 mm	1.937	1.892	−2.33	1.488	−23.18	2.045	5.58	1.470	−24.11
PFRC10	*G_F_* (N/mm)	*G_F_* (N/mm)	∆ (%)	*G_F_* (N/mm)	∆ (%)	*G_F_* (N/mm)	∆ (%)	*G_F_* (N/mm)	∆ (%)
0.5 mm	0.184	0.193	5.22	0.221	20.24	0.202	10.23	0.202	10.23
2.5 mm	1.016	0.994	−2.10	1.206	18.69	1.155	13.76	1.155	13.76
5 mm	2.288	2.212	−3.33	1.844	−19.42	2.485	8.59	1.808	−21.01

## Data Availability

The data presented in this study are available from the authors on request.

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
