# Peer review of "Suitability of Constitutive Models of the Structural Concrete Codes When Applied to Polyolefin Fibre Reinforced Concrete"

_materials, 2022, doi:10.3390/ma15062323_

Round 1

Reviewer 1 Report

Summary:

The paper deals with the constitutive model FRC, i.e. its modification to encompass polyolefin fibres. There is a combination of past and new work, past being experimental and new being simulation, which is beneficial as it combines different aspects of the problem, and a link between research tools is established. The main aim of the paper is to demonstrate the applicability and usefulness of the proposed softening function for the evaluation of complex behaviour that PFRC offers. The main contribution is the evaluation of different models, along with the softening function itself.

Broad comments:

The paper uses mostly clear scientific language in its approach to the subject matter, which is a clear strength; however, improvements are required in certain important aspects of the paper, along with some minor improvements:

  1. The title is informative and relays the subject of the paper, but it does seem to be too long. Authors should think of shortening the title.
  2. It is advisable to avoid keywords that are already included in the title.
  3. The introduction segment of the paper requires editing. This is primarily in regards to that it is not completely known what is the state of the art in the subject area - the literature review is somewhat confusing as it should be better organized with a structured layout of constitutive models. Namely, most of the introduction and attached references concern concrete itself which is not the main focus of the paper: [1-13] concrete in general, [14-28] FRC failure shapes, application and environment. Much of that should be with a focus on constitutive models. Additionally, a major part of previous research information is provided as a group, with no clear analysis of shortcomings of presented papers, which would justify current research need and explain what is being upgraded, i.e. what would be the implications of given research results
  4. Section 2 offers model information, along with some earlier experimental information, thus leaving most of the novelty to Section 3. In order to ensure that section 2 offers solid ground for generalisations and can be viewed as a reliable and robust data source, authors should provide more information on how many series of how many samples provided the basis for the softening function. Statistically sound functions need to be based on a robust data set. It should be emphasized that the information given in section 3 might not be “strong” enough without previously published material.
  5. Since all of the results in section 3 rely on simulated results, the authors need to address certain serious weak points. Namely, it should be elaborated why were the parameters for successful validation selected and what was the accepted error for validation. Additionally, the problems that need to be addressed are
    • Mesh size is one of the more influential parameters on the quality of numerical modelling results – efficiency, convergence, accuracy. The authors do not mention what was the mesh size. Was a sensitivity study done to determine the adopted mesh sizes?
    • The authors should provide more details on how they defined contact interfaces.
  6. In section 5 the authors should mention the implications of this study and its limitations. Limitations could be a basis for future research.

Specific comments:

  1. In references, authors should include DOI numbers where available.

Author Response

Dear Editor,

Please, find enclosed the revised version of the following paper:

 Journal:                     Materials

Manuscript ID:         materials-1609856

Title:                          Suitability of constitutive models of the structural concrete codes
when applied to polyolefin fibre reinforced concrete

Author:                      Alejandro Enfedaque, Fernando Suárez, Marcos G. Alberti, Jaime C. Gálvez

We would like to sincerely thank the reviewers for their comments, which have helped us to improve the quality of the paper, we have found their comments helpful and constructive.

Below we address each of the remarks made by the authors (in blue) to the comments of the reviewers and reference the changes made in the manuscript (highlighted in yellow in the new version).

Yours faithfully,

The authors

Summary:

The paper deals with the constitutive model FRC, i.e. its modification to encompass polyolefin fibres. There is a combination of past and new work, past being experimental and new being simulation, which is beneficial as it combines different aspects of the problem, and a link between research tools is established. The main aim of the paper is to demonstrate the applicability and usefulness of the proposed softening function for the evaluation of complex behaviour that PFRC offers. The main contribution is the evaluation of different models, along with the softening function itself.

Broad comments:

The paper uses mostly clear scientific language in its approach to the subject matter, which is a clear strength; however, improvements are required in certain important aspects of the paper, along with some minor improvements:

  1. The title is informative and relays the subject of the paper, but it does seem to be too long. Authors should think of shortening the title.

Thank you for this suggestion, we have revised the title and think that not many changes can be made, but have slightly modified it as follows:

Suitability of constitutive models of the structural codes when applied to polyolefin fibre reinforced concrete”

  1. It is advisable to avoid keywords that are already included in the title. .

Keyword “polyolefin fibres” has been replaced by “polymer fibres”.

  1. The introduction segment of the paper requires editing. This is primarily in regards to that it is not completely known what is the state of the art in the subject area - the literature review is somewhat confusing as it should be better organized with a structured layout of constitutive models. Namely, most of the introduction and attached references concern concrete itself which is not the main focus of the paper: [1-13] concrete in general, [14-28] FRC failure shapes, application and environment. Much of that should be with a focus on constitutive models. Additionally, a major part of previous research information is provided as a group, with no clear analysis of shortcomings of presented papers, which would justify current research need and explain what is being upgraded, i.e. what would be the implications of given research results.

The authors thank the reviewer for the comment and consequently have carried out a complete revision of the introduction. The final wording can be seen in the following lines.

Concrete has been one of the most employed construction material [1] in modern societies in spite of its impact on the environment [2]. Such use has been based on its resistance to natural weather conditions [3, 4] and its remarkable compressive strength and modulus of elasticity [5, 6]. However, plain concrete is also known as a quasi-brittle material which is not capable of sustaining high-tensile stresses. Adding reinforcing bars to concrete to form reinforced concrete was one of the possible solutions to this problem. Another possibility is adding fibres randomly distributed while mixing forming fibre reinforced concrete (FRC). The effect of the addition of fibres in the mechanical properties of concrete depends on numerous parameters, such as the type of fibres used, dosage, shape and size, distribution, orientation and, among others, pull-out response [7, 8]. Adding certain types of fibres in a determined volume fraction to concrete a material suitable for numerous applications can be obtained. For instance, steel fibre reinforced concrete (SFRC) has been successfully applied to pavements [ 9, 10], tunnel linings [11, 12] or even seismic structures [13, 14]. However, the use of SFRC might have two main drawbacks, its durability under certain circumstances [15, 16] and its environmental impact [17]. One possibility to overcome such durability issues is to use new polymer macro-structural fibres, which are chemically stable when applied in foundations, tunnel linings or marine environments [18, 19, 20, 21]. Polyolefin fibre reinforced concrete (PFRC) is one of the materials available for these uses.

All the previous applications have been possible due to several national codes and recommendations [22, 23, 24, 25, 26, 27]. These documents have set the conditions that a certain FRC should meet so that the improvements provided by the fibres added might enable a reduction in the amount of steel reinforcing bars used. Such requirements take into account the residual strength of the FRC studied at certain crack openings which are related to the service limit state (SLS) and the ultimate limit state (ULS). For checking if a FRC formulation boasts sufficient residual strength, bending tests are performed on FRC specimens in laboratory conditions. Once such tests have been performed, and if the FRC properties exceed the requirements set, the contribution of the fibres might be included in the structural design of the concrete element. Although it is true that both SFRC and PFRC have been shown as suitable for their application in structural concrete elements [7, 28, 29] it should not be overlooked that the extensive experience related with SFRC served as reference when the requirements of the material were set. Once such requirements have been checked, the aforementioned recommendations propose several constitutive models that should be applied in the structural design and that have been successfully applied when using SFRC. Similarly, the constitutive models proposed used several characteristics that were inspired by the mechanical behaviour of SFRC. Consequently, the accuracy of such approaches when dealing with PFRC is a matter that deserves being studied.

The behaviour of PFRC can meet the requirements in the standards [30] although the shapes of the fracture curves have different features compared with those of SFRC [29]. According to the Model Code 2010 (MC2010) [31], the first condition that FRC should meet is a minimum load value, proportional to the maximum strength in the limit of proportionality, at a crack mouth opening displacement of 0.5 mm; this limitation is intended to prevent from a brittle behaviour of the material. Although PFRC with an average fibre dosage might meet the cited requirement, polyolefin fibres need greater deformations to absorb the load that concrete is not capable of sustaining being the first unloading branch steeper than in SFRC. The second condition that MC2010 establishes is that at a crack mouth opening displacement of 2.5 mm the load must be over 20% of the strength at the limit of proportionality. This limitation for PFRC is met even for very low fibre dosages [30]. As there are remarkable differences in the post peak mechanical behaviours of SFRC and PFRC, the applicability of the constitutive models, which are built based on certain values of their residual strength, should be studied. Thus, this study seeks to address the main issues concerning the considerations in the standards regarding the mechanical behaviour of PFRC.

The main goals of this contribution are the following. Firstly, an implementation of the linear and rigid-plastic models developed in MC2010 in a numerical code by means of a material subroutine is sought. Secondly, the differences between the material behaviour obtained with the constitutive models of MC2010 and those obtained using the trilinear approach proposed by the authors that delivers an accurate reproduction of the PFRC behaviour will be assessed. This matter will be carried out in four mixes of PFRC with 3, 4.5, 6 and 10 kg/m³ of polyolefin fibres. Lastly, an evaluation of the material behaviour at SLS and ULS and another regarding the fracture energy consumed in the tests will be carried out for determining the reliability of using the studied approaches. Summarising, this manuscript supplies additional information for a possible future adaptation of MC2010, and other national concrete structural codes based on it when applied to PFRC.

  1. Section 2 offers model information, along with some earlier experimental information, thus leaving most of the novelty to Section 3. In order to ensure that section 2 offers solid ground for generalisations and can be viewed as a reliable and robust data source, authors should provide more information on how many series of how many samples provided the basis for the softening function. Statistically sound functions need to be based on a robust data set. It should be emphasized that the information given in section 3 might not be “strong” enough without previously published material. .

This paragraph has been modified to include some information about the experimental campaign used for reference, as suggested. The text now reads:

“The reference experimental results correspond to four PFRC mixes of VCC with four fibre dosages: 3, 4.5, 6 and 10 kg/m3. These are selected results from a broader experimental campaign that studied several PFRC mixes and analysed their mechanical properties through a series of two prismatic and nine cylindrical specimens of each mix; the details regarding these experimental works may be consulted in [28, 29].”

  1. Since all of the results in section 3 rely on simulated results, the authors need to address certain serious weak points. Namely, it should be elaborated why were the parameters for successful validation selected and what was the accepted error for validation. Additionally, the problems that need to be addressed are
  • Mesh size is one of the more influential parameters on the quality of numerical modelling results – efficiency, convergence, accuracy. The authors do not mention what was the mesh size. Was a sensitivity study done to determine the adopted mesh sizes?
  • The authors should provide more details on how they defined contact interfaces.

Regarding the first part of this remark, the equations described in MC2010 that allow defining the rigid-plastic and linear models are presented in section “2.2 Model adopted by the MC2010” in equations (8)-(10), but it is true that the text was not clear enough on what values were used for each model. The following text has been added immediately before Table 1:

“These diagrams have been defined following the MC2010, thus a limit value of the crack opening of 2.5 mm has been employed and equations (8)-(10) used with the values of fR1 and fR3 experimentally obtained, which can be consulted in [28, 29].”

Regarding the mesh, it was refined based on previous numerical simulations with similar PFRC mixes, in which the mesh dependence was already analysed. To explain this, the following text has been included (which entails including a new reference):

“…; this refinement was designed based on previous works (see [37] and [38]) where the mesh dependence was already analysed.”

As for the contact interfaces, no contact interfaces have been defined in the model, since fracture is modelled as an embedded crack that can develop in any element of the model, thus there is no need of defining cracking surfaces.

  1. In section 5 the authors should mention the implications of this study and its limitations. Limitations could be a basis for future research. .

The authors have modified the conclusions in order to highlight the most important findings and also to point out the limitations of the research. The final wording can be seen in the following lines.

This contribution has implemented the constitutive relations proposed by the MC2010 for FRC in a user material subroutine seeking to assess the adaptability of these constitutive relations to PFRC. In addition, a trilinear model and an unlimited version of the linear model proposed by MC2010 have been employed as comparisons.

The simulations carried out with the implementation of the trilinear softening function proposed by the authors showed a remarkable resemblance with the experimental behaviour of PFRC. Regarding the simulations performed through using the constitutive models proposed by MC2010, it was observed that they might not replicate the flexural behaviour of PFRC precisely. Neither the peak load nor the unloading process appeared in the simulations performed with the rigid-plastic and linear constitutive relations.

When analysing the load values obtained at 0.5mm of deflection, it is seen that the constitutive relations proposed by the MC2010 are above the experimental results. At ULS, the relations proposed by the MC2010 and that proposed by the authors are able to reproduce with notable exactness the experimental load values. Regarding the ultimate load-bearing capacity of the material, which was assumed at a deflection of 5mm, both the trilinear and unlimited linear model were close to the load-bearing capacity of the material. On the contrary, the rigid-plastic model and the limited version of the linear model clearly underestimated the mechanical properties of PFRC.

Based on the aforementioned arguments, it seems that the rigid-plastic and the linear models proposed by the MC2010 might be employed with caution at SLS when applied to PFRC. This might be a result of the limited degree of accuracy that the idealisation considered in MC2010 may offer when applied to FRC formulations where fR3 is notably greater than fR1. The authors consider that if the results of the EN-14651 fracture tests boast a clear three-stretch behaviour after reaching the peak load (unloading-reloading-final unloading), a trilinear constitutive relation might be adopted as an alternative to the models proposed by MC2010. This is relevant given that the latter models were developed considering that steel fibres were the main option for reinforcing concrete with fibres.

The limitation of this study might be that the differences detected where determined for certain constitutive models stated in MC2010 and consequently if such models are modified the present conclusions should be re-evaluated.

Specific comments:

  1. In references, authors should include DOI numbers where available.

The authors thank the comment of the reviewer as such data will help to understand the state of the art of the matter. Consequently, such information has been included in the references.

Reviewer 2 Report

1. Why is the word concrete 2 times in the title?
2. Why is there no numerical result in the abstract of the technical manuscript?
3. The beginning of the introduction is unsuccessful. It's not scientific to write "some time"
4. Literary review is rather one-sided. This highly rated journal should analyze building materials based on articles from recent years. For instance:
- A D Tolstoy; V S Lesovik; E S Glagolev; A I Krymova. Synergetics of hardening construction systems. IOP Conference Series: Materials Science and Engineering. 2018. 327(3), 032056. doi: 10.1088/1757-899X/327/3/032056
- Amran, Mugahed; Murali, G.; Khalid, Nur Hafizah A.; Fediuk, Roman; Ozbakkaloglu, Togay; Lee, Yeong Huei; Haruna, Sani; Lee, Yee Yong. Slag uses in making an ecofriendly and sustainable concrete: A review. Construction and Building Materials, 2021. doi.org/10.1016/J.CONBUILDMAT.2020.121942
5. At the end of the introduction, the goal and objectives for achieving it should be clearly indicated. And in the conclusions, give a numbered list of findings corresponding to the tasks
6. The scientific contribution of this article is not completely clear. Do you want to make a change to the codes?

Author Response

We would like to sincerely thank the reviewers for their comments, which have helped us to improve the quality of the paper, we have found their comments helpful and constructive.

Below we address each of the remarks made by the authors (in blue) to the comments of the reviewers and reference the changes made in the manuscript (highlighted in yellow in the new version).

Yours faithfully,

The authors

1. Why is the word concrete 2 times in the title?

The title has been slightly modified and now reads:

Suitability of constitutive models of the structural codes when applied to polyolefin fibre reinforced concrete”

2. Why is there no numerical result in the abstract of the technical manuscript?

The abstract mentioned the numerical simulations but, in order to more explicitly present the main findings of the work, the additional text has been included:

“…The results show remarkable differences between the experimental results and the numerical simulations when the constitutive models described in the MC2010 are employed for different polyolefin fibre reinforced concrete mixes, while the material behaviour can be reproduced with greater accuracy if the softening function proposed by the authors is employed when this type of macro-polymer fibres is used.…”

3. The beginning of the introduction is unsuccessful. It's not scientific to write "some time"

The authors thank comment 3, 4 and 5 of the reviewer and in order to answer all of them the introduction have been re-written. Some of the previous information has been deleted trying to focus  on the contributions of the manuscript. The final wording can be seen in the following lines.

Concrete has been one of the most employed construction material [1] in modern societies in spite of its impact on the environment [2]. Such use has been based on its resistance to natural weather conditions [3, 4] and its remarkable compressive strength and modulus of elasticity [5, 6]. However, plain concrete is also known as a quasi-brittle material which is not capable of sustaining high-tensile stresses. Adding reinforcing bars to concrete to form reinforced concrete was one of the possible solutions to this problem. Another possibility is adding fibres randomly distributed while mixing forming fibre reinforced concrete (FRC). The effect of the addition of fibres in the mechanical properties of concrete depends on numerous parameters, such as the type of fibres used, dosage, shape and size, distribution, orientation and, among others, pull-out response [7, 8]. Adding certain types of fibres in a determined volume fraction to concrete a material suitable for numerous applications can be obtained. For instance, steel fibre reinforced concrete (SFRC) has been successfully applied to pavements [ 9, 10], tunnel linings [11, 12] or even seismic structures [13, 14]. However, the use of SFRC might have two main drawbacks, its durability under certain circumstances [15, 16] and its environmental impact [17]. One possibility to overcome such durability issues is to use new polymer macro-structural fibres, which are chemically stable when applied in foundations, tunnel linings or marine environments [18, 19, 20, 21]. Polyolefin fibre reinforced concrete (PFRC) is one of the materials available for these uses.

All the previous applications have been possible due to several national codes and recommendations [22, 23, 24, 25, 26, 27]. These documents have set the conditions that a certain FRC should meet so that the improvements provided by the fibres added might enable a reduction in the amount of steel reinforcing bars used. Such requirements take into account the residual strength of the FRC studied at certain crack openings which are related to the service limit state (SLS) and the ultimate limit state (ULS). For checking if a FRC formulation boasts sufficient residual strength, bending tests are performed on FRC specimens in laboratory conditions. Once such tests have been performed, and if the FRC properties exceed the requirements set, the contribution of the fibres might be included in the structural design of the concrete element. Although it is true that both SFRC and PFRC have been shown as suitable for their application in structural concrete elements [7, 28, 29] it should not be overlooked that the extensive experience related with SFRC served as reference when the requirements of the material were set. Once such requirements have been checked, the aforementioned recommendations propose several constitutive models that should be applied in the structural design and that have been successfully applied when using SFRC. Similarly, the constitutive models proposed used several characteristics that were inspired by the mechanical behaviour of SFRC. Consequently, the accuracy of such approaches when dealing with PFRC is a matter that deserves being studied.

The behaviour of PFRC can meet the requirements in the standards [30] although the shapes of the fracture curves have different features compared with those of SFRC [29]. According to the Model Code 2010 (MC2010) [31], the first condition that FRC should meet is a minimum load value, proportional to the maximum strength in the limit of proportionality, at a crack mouth opening displacement of 0.5 mm; this limitation is intended to prevent from a brittle behaviour of the material. Although PFRC with an average fibre dosage might meet the cited requirement, polyolefin fibres need greater deformations to absorb the load that concrete is not capable of sustaining being the first unloading branch steeper than in SFRC. The second condition that MC2010 establishes is that at a crack mouth opening displacement of 2.5 mm the load must be over 20% of the strength at the limit of proportionality. This limitation for PFRC is met even for very low fibre dosages [30]. As there are remarkable differences in the post peak mechanical behaviours of SFRC and PFRC, the applicability of the constitutive models, which are built based on certain values of their residual strength, should be studied. Thus, this study seeks to address the main issues concerning the considerations in the standards regarding the mechanical behaviour of PFRC.

The main goals of this contribution are the following. Firstly, an implementation of the linear and rigid-plastic models developed in MC2010 in a numerical code by means of a material subroutine is sought. Secondly, the differences between the material behaviour obtained with the constitutive models of MC2010 and those obtained using the trilinear approach proposed by the authors that delivers an accurate reproduction of the PFRC behaviour will be assessed. This matter will be carried out in four mixes of PFRC with 3, 4.5, 6 and 10 kg/m³ of polyolefin fibres. Lastly, an evaluation of the material behaviour at SLS and ULS and another regarding the fracture energy consumed in the tests will be carried out for determining the reliability of using the studied approaches. Summarising, this manuscript supplies additional information for a possible future adaptation of MC2010, and other national concrete structural codes based on it when applied to PFRC.

4. Literary review is rather one-sided. This highly rated journal should analyze building materials based on articles from recent years. For instance:

- A D Tolstoy; V S Lesovik; E S Glagolev; A I Krymova. Synergetics of hardening construction systems. IOP Conference Series: Materials Science and Engineering. 2018. 327(3), 032056. doi: 10.1088/1757-899X/327/3/032056

- Amran, Mugahed; Murali, G.; Khalid, Nur Hafizah A.; Fediuk, Roman; Ozbakkaloglu, Togay; Lee, Yeong Huei; Haruna, Sani; Lee, Yee Yong. Slag uses in making an ecofriendly and sustainable concrete: A review. Construction and Building Materials, 2021. doi.org/10.1016/J.CONBUILDMAT.2020.121942

The authors have included the suggestions in the new version of the introduction shown in comment 3.

5. At the end of the introduction, the goal and objectives for achieving it should be clearly indicated. And in the conclusions, give a numbered list of findings corresponding to the tasks

The authors have included the suggestions in the new version of the introduction shown in comment 3.

6. The scientific contribution of this article is not completely clear. Do you want to make a change to the codes?

The authors would like to remark that this contribution is only confronting how the approach stated in certain codes fit with the mechanical behaviour of polyolefin fibre reinforced concrete. Although the authors reckon that there are differences that cannot be overlooked, it should not be forgotten that the recommendations of the structural codes are based on a thorough study of numerous analytical, numerical and experimental cases. Consequently, the findings that are shown in this paper, if confirmed by other researchers, might question if a unique structural approach is valid for all types of fibre reinforced concrete.

Reviewer 3 Report

This is an interesting manuscript that seeks to address the main issues concerning the considerations in the standards of the mechanical behaviour of Polyolefin fibre reinforced concrete.

The paper is well structured and clearly written.

Although the English language used in the article is not bad, a revision would be advisable because the manuscript is difficult to follow.

Descriptions of the references should be extended. For example in a sentence “The properties that steel fibres confer to concrete when forming steel fibre reinforced concrete (SFRC) have enabled use of such a material in a wide range of applications, such as pavements, [16, 17] tunnel linings [18, 19] or even seismic structures [20, 21].” there are 6 references, but the reader does not know what they refer to.

Check the quality of the figure 9 because it is very difficult to read the text.

Conclusions should be rewritten to clearly highlight the main findings of the paper.

From these reasons, I think that this manuscript, after a minor revision, could be accepted to be published by the Materials journal.

Author Response

We would like to sincerely thank the reviewers for their comments, which have helped us to improve the quality of the paper, we have found their comments helpful and constructive.

Below we address each of the remarks made by the authors (in blue) to the comments of the reviewers and reference the changes made in the manuscript (highlighted in yellow in the new version).

Yours faithfully,

The authors

This is an interesting manuscript that seeks to address the main issues concerning the considerations in the standards of the mechanical behaviour of Polyolefin fibre reinforced concrete.

The paper is well structured and clearly written.

Although the English language used in the article is not bad, a revision would be advisable because the manuscript is difficult to follow.

Descriptions of the references should be extended. For example in a sentence “The properties that steel fibres confer to concrete when forming steel fibre reinforced concrete (SFRC) have enabled use of such a material in a wide range of applications, such as pavements, [16, 17] tunnel linings [18, 19] or even seismic structures [20, 21].” there are 6 references, but the reader does not know what they refer to.

The introduction of the manuscript has been completely re-written in order to clarify the state of the art and the soundness of the research. The final wording can be seen in the following lines.

Concrete has been one of the most employed construction material [1] in modern societies in spite of its impact on the environment [2]. Such use has been based on its resistance to natural weather conditions [3, 4] and its remarkable compressive strength and modulus of elasticity [5, 6]. However, plain concrete is also known as a quasi-brittle material which is not capable of sustaining high-tensile stresses. Adding reinforcing bars to concrete to form reinforced concrete was one of the possible solutions to this problem. Another possibility is adding fibres randomly distributed while mixing forming fibre reinforced concrete (FRC). The effect of the addition of fibres in the mechanical properties of concrete depends on numerous parameters, such as the type of fibres used, dosage, shape and size, distribution, orientation and, among others, pull-out response [7, 8]. Adding certain types of fibres in a determined volume fraction to concrete a material suitable for numerous applications can be obtained. For instance, steel fibre reinforced concrete (SFRC) has been successfully applied to pavements [ 9, 10], tunnel linings [11, 12] or even seismic structures [13, 14]. However, the use of SFRC might have two main drawbacks, its durability under certain circumstances [15, 16] and its environmental impact [17]. One possibility to overcome such durability issues is to use new polymer macro-structural fibres, which are chemically stable when applied in foundations, tunnel linings or marine environments [18, 19, 20, 21]. Polyolefin fibre reinforced concrete (PFRC) is one of the materials available for these uses.

All the previous applications have been possible due to several national codes and recommendations [22, 23, 24, 25, 26, 27]. These documents have set the conditions that a certain FRC should meet so that the improvements provided by the fibres added might enable a reduction in the amount of steel reinforcing bars used. Such requirements take into account the residual strength of the FRC studied at certain crack openings which are related to the service limit state (SLS) and the ultimate limit state (ULS). For checking if a FRC formulation boasts sufficient residual strength, bending tests are performed on FRC specimens in laboratory conditions. Once such tests have been performed, and if the FRC properties exceed the requirements set, the contribution of the fibres might be included in the structural design of the concrete element. Although it is true that both SFRC and PFRC have been shown as suitable for their application in structural concrete elements [7, 28, 29] it should not be overlooked that the extensive experience related with SFRC served as reference when the requirements of the material were set. Once such requirements have been checked, the aforementioned recommendations propose several constitutive models that should be applied in the structural design and that have been successfully applied when using SFRC. Similarly, the constitutive models proposed used several characteristics that were inspired by the mechanical behaviour of SFRC. Consequently, the accuracy of such approaches when dealing with PFRC is a matter that deserves being studied.

The behaviour of PFRC can meet the requirements in the standards [30] although the shapes of the fracture curves have different features compared with those of SFRC [29]. According to the Model Code 2010 (MC2010) [31], the first condition that FRC should meet is a minimum load value, proportional to the maximum strength in the limit of proportionality, at a crack mouth opening displacement of 0.5 mm; this limitation is intended to prevent from a brittle behaviour of the material. Although PFRC with an average fibre dosage might meet the cited requirement, polyolefin fibres need greater deformations to absorb the load that concrete is not capable of sustaining being the first unloading branch steeper than in SFRC. The second condition that MC2010 establishes is that at a crack mouth opening displacement of 2.5 mm the load must be over 20% of the strength at the limit of proportionality. This limitation for PFRC is met even for very low fibre dosages [30]. As there are remarkable differences in the post peak mechanical behaviours of SFRC and PFRC, the applicability of the constitutive models, which are built based on certain values of their residual strength, should be studied. Thus, this study seeks to address the main issues concerning the considerations in the standards regarding the mechanical behaviour of PFRC.

The main goals of this contribution are the following. Firstly, an implementation of the linear and rigid-plastic models developed in MC2010 in a numerical code by means of a material subroutine is sought. Secondly, the differences between the material behaviour obtained with the constitutive models of MC2010 and those obtained using the trilinear approach proposed by the authors that delivers an accurate reproduction of the PFRC behaviour will be assessed. This matter will be carried out in four mixes of PFRC with 3, 4.5, 6 and 10 kg/m³ of polyolefin fibres. Lastly, an evaluation of the material behaviour at SLS and ULS and another regarding the fracture energy consumed in the tests will be carried out for determining the reliability of using the studied approaches. Summarising, this manuscript supplies additional information for a possible future adaptation of MC2010, and other national concrete structural codes based on it when applied to PFRC.

Check the quality of the figure 9 because it is very difficult to read the text.

The authors thank the comment of the reviewer and have improved the quality of Figure 9.

Conclusions should be rewritten to clearly highlight the main findings of the paper.

The authors appreciate the comment of the reviewer as a more concise section might result in a greater impact of the conclusions. The following paragraphs show the final wording of such section.

This contribution has implemented the constitutive relations proposed by the MC2010 for FRC in a user material subroutine seeking to assess the adaptability of these constitutive relations to PFRC. In addition, a trilinear model and an unlimited version of the linear model proposed by MC2010 have been employed as comparisons.

The simulations carried out with the implementation of the trilinear softening function proposed by the authors showed a remarkable resemblance with the experimental behaviour of PFRC. Regarding the simulations performed through using the constitutive models proposed by MC2010, it was observed that they might not replicate the flexural behaviour of PFRC precisely. Neither the peak load nor the unloading process appeared in the simulations performed with the rigid-plastic and linear constitutive relations.

When analysing the load values obtained at 0.5mm of deflection, it is seen that the constitutive relations proposed by the MC2010 are above the experimental results. At ULS, the relations proposed by the MC2010 and that proposed by the authors are able to reproduce with notable exactness the experimental load values. Regarding the ultimate load-bearing capacity of the material, which was assumed at a deflection of 5mm, both the trilinear and unlimited linear model were close to the load-bearing capacity of the material. On the contrary, the rigid-plastic model and the limited version of the linear model clearly underestimated the mechanical properties of PFRC.

Based on the aforementioned arguments, it seems that the rigid-plastic and the linear models proposed by the MC2010 might be employed with caution at SLS when applied to PFRC. This might be a result of the limited degree of accuracy that the idealisation considered in MC2010 may offer when applied to FRC formulations where fR3 is notably greater than fR1. The authors consider that if the results of the EN-14651 fracture tests boast a clear three-stretch behaviour after reaching the peak load (unloading-reloading-final unloading), a trilinear constitutive relation might be adopted as an alternative to the models proposed by MC2010. This is relevant given that the latter models were developed considering that steel fibres were the main option for reinforcing concrete with fibres.

The limitation of this study might be that the differences detected where determined for certain constitutive models stated in MC2010 and consequently if such models are modified the present conclusions should be re-evaluated.

From these reasons, I think that this manuscript, after a minor revision, could be accepted to be published by the Materials journal.

Round 2

Reviewer 2 Report

good revision!